# Multiple alleles at a single locus control seed dormancy in Swedish *Arabidopsis*

**Envel Kerdaffrec, Danièle L Filiault, Arthur Korte, Eriko Sasaki, Viktoria Nizhynska, Ümit Seren, Magnus Nordborg\***

Gregor Mendel Institute, Austrian Academy of Sciences, Vienna Biocenter, Vienna, Austria

**Abstract** Seed dormancy is a complex life history trait that determines the timing of germination and is crucial for local adaptation. Genetic studies of dormancy are challenging, because the trait is highly plastic and strongly influenced by the maternal environment. Using a combination of statistical and experimental approaches, we show that multiple alleles at the previously identified dormancy locus *DELAY OF GERMINATION1* jointly explain as much as 57% of the variation observed in Swedish *Arabidopsis thaliana*, but give rise to spurious associations that seriously mislead genome-wide association studies unless modeled correctly. Field experiments confirm that the major alleles affect germination as well as survival under natural conditions, and demonstrate that locally adaptive traits can sometimes be dissected genetically.

## Introduction

The genetic basis of complex trait variation is a major challenge for modern biology. There are long-standing debates over whether alleles are common or rare, loci few or many, effects small or large, interactions additive or epistatic, and so on—questions of practical importance for agriculture and human health, and of fundamental importance for evolutionary biology. Lately, the discourse has been heavily influenced by the mass of human genome-wide association studies (GWAS), which have generally revealed a highly polygenic basis for complex traits, with numerous causative polymorphisms of small effect distributed across the genome (*Visscher et al., 2012*). However, the relevance of these studies for understanding adaptive variation is not clear, as they have almost exclusively involved traits likely to have been under stabilizing or purifying selection. Indeed, among the few GWAS of adaptive variation that have been carried out, some have identified common alleles of major effect (including for human traits such as pigmentation) (*Atwell et al., 2010*; *Beleza et al., 2013*).

Here we consider seed dormancy, a complex life history trait that plays a major role in local adaptation for many plant species (*Finch-Savage and Leubner-Metzger, 2006*). For a weedy annual like *Arabidopsis thaliana*, the subject of our study, germinating at the right time is crucial for early survival, and also influences other life history traits, such as flowering (*Postma and Ågren, 2016*; *Huang et al., 2010a*; *Chiang et al., 2013*; *Donohue et al., 2010*; *Springthorpe and Penfield, 2015*). Seed dormancy varies geographically, with high dormancy being associated with long, dry summers, and low dormancy being more prevalent where summers are short and wet (*Chiang et al., 2011*; *Debieu et al., 2013*; *Kronholm et al., 2012*). Analysis of a diallel cross between a highly dormant line from the Cape Verde Islands and a non-dormant line from Poland revealed several quantitative trait loci (QTL), one of which, *DELAY OF GERMINATION1* (*DOG1*), has been cloned and molecularly characterized (*Alonso-Blanco et al., 2003*; *Bentsink et al., 2006*), although its function is poorly understood. It appears to be tightly regulated by a complex array of mechanisms that include alternative splicing, alternative polyadenylation, and a cis-acting antisense non-coding

**\*For correspondence:** magnus. nordborg@gmi.oeaw.ac.at

transcript (*asDOG1*) (*Bentsink et al., 2006*; *Nakabayashi et al., 2015*; *Cyrek et al., 2016*; *Fedak et al., 2016*), but, although *DOG1* mRNA and protein abundance generally correlate with seed dormancy, this relationship is not observed in all accessions (*Bentsink et al., 2006*; *Chiang et al., 2011*; *Nakabayashi et al., 2012*; *Fedak et al., 2016*). In these cases, epistatic interactions with the genetic background, or other mechanisms, such as the critical self-dimerization of the DOG1 protein, might come into play (*Nakabayashi et al., 2015*). The *DOG1* locus harbors a series of functionally distinct alleles segregating in wild populations, and exhibits a pattern of genetic differentiation suggestive of local adaptation (*Kronholm et al., 2012*). Moreover, field experiments using a diallel cross between accessions from Sweden and Italy identified a major QTL associated with total fitness that spanned the *DOG1* region, and also demonstrated that genetic differences in seedling establishment—and most particularly in seed dormancy—are one of the main components of local adaptation (*Postma and Ågren, 2016*; *Postma et al., 2016*; *Huang et al., 2010a*). Finally, extensive natural variation for seed dormancy is not unique to the *DOG1* locus and many other QTLs have been mapped in various populations (*Alonso-Blanco et al., 2003*; *Bentsink et al., 2006*, *2010*; *Amiguet-Vercher et al., 2015*; *Xiang et al., 2016*). The recent cloning of the *DELAY OF GER-MINATION18* (*DOG18*)/*IBO* QTLs in two independent diallel crosses revealed the existence of numerous loss-of-function alleles—mainly caused by nonsense mutations and structural variants—at the *REDUCED DORMANCY5* (*RDO5*) locus in wild *A. thaliana* populations (*Xiang et al., 2016*; *Amiguet-Vercher et al., 2015*; *Xiang et al., 2014*).

## Results and discussion

To investigate the genetic basis for seed dormancy variation in a natural population, we carried out a GWAS using 161 lines from Sweden (*Figure 1—source data 1*), where seeds are normally set in early summer and seedling establishment mostly occurs in fall. We focused on the germination rate of seeds after-ripened for 21 days (GR21), which showed dramatic variation with 46% of lines being dormant (GR21 < 20%) and 35% being effectively non-dormant (GR21 > 80%) (*Figure 1A*). We observed a strong correlation between GR21 and latitude (Pearson's $r$ = 0.62) (*Figure 1B*), largely due to the fact that northern Swedish lines were mostly non-dormant whereas the majority of southern Swedish lines was dormant (*Figure 1C* and online: https://goo.gl/3HeqX5). GR21 was also strongly correlated with both temperature and length of growing season, consistent with genetic variation for seed dormancy being adaptive (*Supplementary file 1*).

GWAS for GR21 identified a single highly significant peak spanning almost 50 kb at about 18.6 Mb on chromosome 5 (*Figure 2A* and online: https://goo.gl/30EPt3). The region includes *DOG1*, but the significantly associated SNPs were located 21 kb away from this obvious candidate, and several other genes in the region could possibly be involved in dormancy (*Figure 2B and D*, *Figure 2—figure supplement 1* and Materials and methods). Analysis of tDNA insertion lines (in the Col-0 background) for three of these genes revealed that *CIPK20*—the first gene downstream of *DOG1*—could potentially affect dormancy as well (although it could also be the case that the *cipk20* tDNA insertion directly affects regulation of *DOG1*; see *Figure 2—figure supplement 1*). Thus, although *DOG1* seemed an excellent candidate for explaining dormancy variation (*Alonso-Blanco et al., 2003*; *Bentsink et al., 2006*, *2010*; *Kronholm et al., 2012*), we could not exclude that neighbouring genes also contribute.

However, a simpler reason for the distant association is that allelic heterogeneity at the *DOG1* locus (previously described by *Kronholm et al., 2012*) gives rise to 'synthetic', or 'ghost' associations in the region. This phenomenon, which is well-documented in both *A. thaliana* (*Atwell et al., 2010*) and rice (*Huang et al., 2010b*), is a simple statistical artifact that arises from fitting a single-locus, two-allele model when the true cause of phenotypic variation is multiple correlated factors (*Atwell et al., 2010*; *Dickson et al., 2010*; *Platt et al., 2010*).

To look for evidence of allelic heterogeneity and synthetic associations, we examined linkage disequilibrium (LD) between the most strongly associated SNP (SNP1, $-\log_{10}p$ = 8.73) and the other markers in the region. As is typical in *A. thaliana* (*Kim et al., 2007*; *Long et al., 2013*), LD decayed within 10 kb of SNP1, however, we identified one marker, SNP2, in moderate LD ($r^2$ = 0.61) with SNP1 despite a separation of almost 21 kb (*Figure 2—figure supplement 2*). SNP2 was strongly, albeit non-significantly, associated with GR21 ($-\log_{10}p$ = 5.80) and is located in the promoter region of *DOG1*. As *DOG1* expression is often correlated with seed dormancy (*Bentsink et al., 2006*;

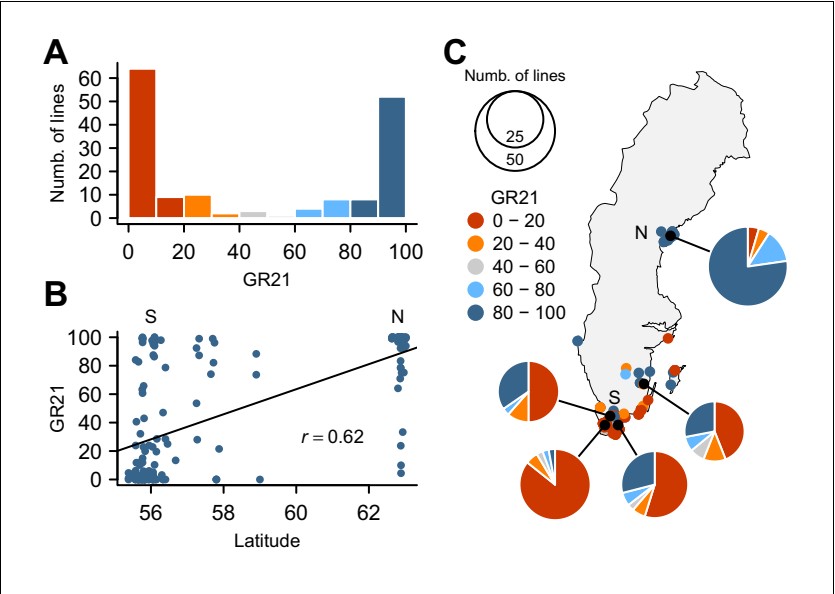

**Figure 1.** Natural variation for seed dormancy in Swedish *A. thaliana*. (**A**) The distribution of the germination rate 21 days after seed harvest (GR21). (**B**) Correlation between GR21 and latitude. We define southern Sweden (S) as the region below 58°N and northern Sweden (N) as above 62°N. (**C**) Geographic distribution of GR21. An interactive version of this map is available online (https://goo.gl/3HeqX5).

The following source data is available for figure 1:

**Source data 1.** The 161 Swedish accessions used in this study.

---

*Chiang et al., 2011*; *Nakabayashi et al., 2012*; *Fedak et al., 2016*), we hypothesized that SNP2 could contribute to GR21 variation, and used it as starting point for a multi-locus GWAS based on stepwise inclusion of markers (*Segura et al., 2012*). The optimal model resulting from starting with SNP2 explained 57% of the variation and consisted of two markers in addition to SNP2: SNP3, also in the *DOG1* promoter; and SNP4, a rare allele [minor allele frequency (MAF) = 0.018] in the first exon of *DOG1* (*Figure 2C–D* and *Figure 2—figure supplement 3B and E*). In contrast, the optimal model resulting from starting with SNP1 explained only 46% of the variation, and identified only one additional marker: SNP4 (*Figure 2—figure supplement 3A and D*). Starting from SNP3, on the other hand, yielded the same model as starting from SNP2, and it is clear that the combined effect of SNP2 and SNP3 effectively eliminates the signal at SNP1 (*Figure 2—figure supplement 3B–C and E–F*), suggesting that it (and the small peak around it) is a synthetic association that serves as a proxy for SNP2 and SNP3 (or closely linked variants: SNP3 is in perfect LD with two other nearby SNPs that can explain the variation just as well; see Materials and methods). The rare SNP4, on the other hand, appears to be independent of the other SNPs. In line with these stepwise results, we found that SNP2 and SNP3 were the best pairwise predictors of the dormancy variation (see Materials and methods and *Supplementary file 2*). Taken together, these GWAS results strongly suggest that variants in the *DOG1* promoter region underlie dormancy variation in the Swedish population (but say nothing about the molecular mechanism, a fact to which we will return below).

To confirm the hypotheses generated by GWAS, we turned to transgenics and experimental crosses. Linkage analysis in six different F2 populations confirmed the importance of the *DOG1* region (*Figure 3A* and *Figure 3—source data 1*). Two of the crosses (TFÄ 07 x Stenk-3 and Eden-2 x ÖMö1-7) demonstrated that the dormant SNP2 allele does not require SNP1, as expected if the association observed at the latter is synthetic. However, due to the strong linkage disequilibrium between SNP3 and SNP1, no cross let us similarly demonstrate that the dormant SNP3 has an effect independently of SNP1. To overcome these limitations, we complemented the non-dormant *dog1-2* mutant with genomic fragments—promoter and gene—corresponding to different *DOG1* alleles. Lines transformed with predicted dormant alleles carrying either SNP2 or SNP3 (but not SNP1, which

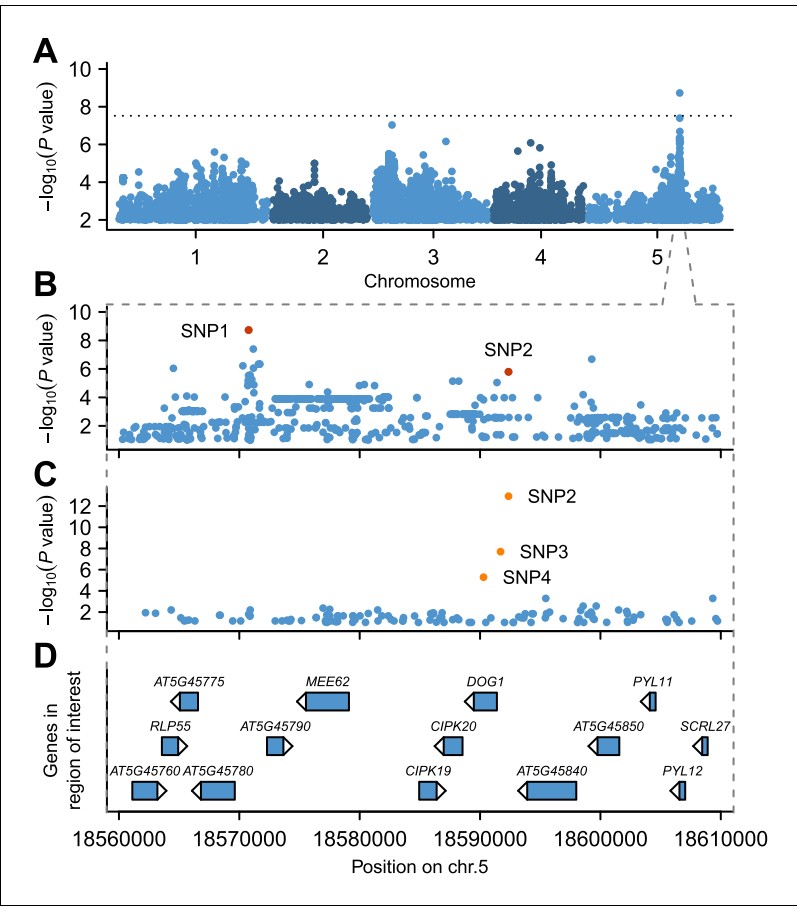

**Figure 2.** GWAS for primary seed dormancy on a set of 161 *A. thaliana* Swedish accessions. (**A**) Manhattan plot of genome-wide association results for GR21. The dotted horizontal line indicates a significance level of 0.05 after Bonferroni correction for multiple testing. (**B**) Magnification of the peak region on chromosome 5. The most significant SNP (SNP1) was located at position 18,570,773. SNP2 (18,592,365) is in moderate LD with SNP1, despite a physical separation of almost 21 kb. Both SNPs are highlighted. (**C**) Output of a local multi-locus association scan using SNP2 as starting point. The resulting optimal model consisted of three highlighted SNPs at the *DOG1* locus: SNP2, SNP3 (18,591,702) and SNP4 (18,590,289). (**D**) Annotated genes in the region under the peak. The GWAS results can be viewed interactively online (https://goo.gl/30EPt3).

The following figure supplements are available for figure 2:

**Figure supplement 1.** The effect of *mee62*, *cipk19* and *cipk20* knock-outs on dormancy.

**Figure supplement 2.** The linkage disequilibrium pattern in the region surrounding *DOG1*.

**Figure supplement 3.** Details of local MLMM association scans for GR21.

is not part of the transformed fragment) were significantly more dormant than those transformed with a predicted non-dormant allele (*Figure 3B* and *Figure 3—source data 2*). It is thus clear that *DOG1* alleles tagged by SNP2 and SNP3 are capable of producing a dormant phenotype independently of SNP1 (and other closely linked genes, like *CIPK20*). Finally, two additional F2 populations (TEDEN 03 x TÅL 07 and TAA 14 x TEDEN 03) supported the association between dormancy and SNP4, thus bringing the number of distinct dormant *DOG1* alleles to three (*Figure 3A*).

In summary, GWAS identified four *DOG1* haplotypes, three associated with dormancy (D2-D4, tagged by the late alleles of SNP2-SNP4, respectively), and one associated with non-dormancy (ND, tagged by early alleles at all three SNP loci) (*Figure 3C*). These SNPs may be causative, but we emphasize that SNP3 is in perfect LD with two other SNPs a few bp away that can explain the

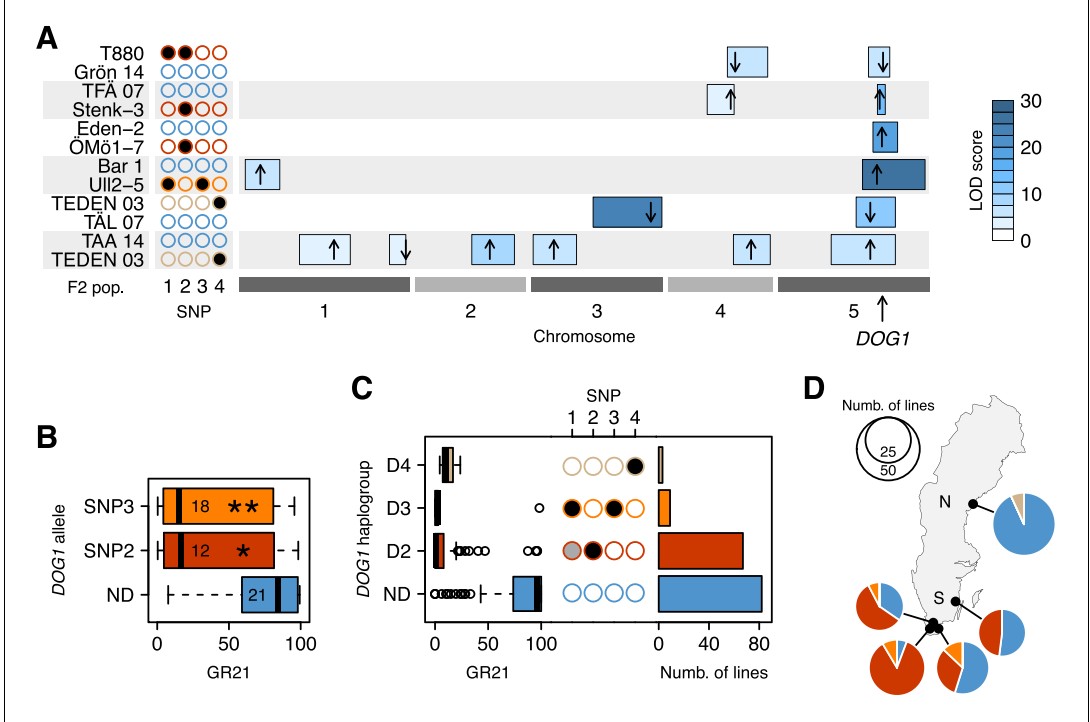

**Figure 3.** Functional validation and geographic distribution of *DOG1* alleles. (**A**) Summary of QTL mapping for GR21 in six different F2 populations. Open and closed circles correspond to the reference (non-dormant) and non-reference (all dormant) alleles at SNP1-SNP4, respectively. Circles are colored as in *Figure 3C*, to represent the haplotypes carried by parental lines. Horizontal boxes filled with shades of blue represent 95% confidence intervals of QTLs along the five *A. thaliana* chromosomes (*Figure 3—source data 1*). The position of the highest LOD score within a given interval is marked with a black arrow, whose direction indicates the allelic effect of the first parent (up increases GR21; down decreases GR21). Only QTLs with a significant LOD score are depicted. (**B**) Transgenic complementation of the non-dormant *dog1-2* mutant with two dormant *DOG1* alleles carrying either SNP2 or SNP3, or a non-dormant allele (ND) as control (*Figure 3—source data 2*). A t-test was performed to compare the effect of the dormant alleles to that of the non-dormant allele (p=0.0147 and p=0.0046 for the alleles tagged by SNP2 and SNP3, respectively). (**C**) GR21 variation within the four *DOG1* haplogroups defined by SNP2-SNP4. The frequency of SNP1-SNP4 within each haplogroup is indicated with open (SNP is absent), grey (SNP segregates) and closed (SNP is fixed) circles. The barplot indicates the haplogroup frequency in the 161 Swedish lines. (**D**) Geographic distribution of *DOG1* haplogroups. As in *Figure 1*, N and S stand for northern and southern Sweden, respectively. An interactive version of this map is available online (https://goo.gl/nfNQjh).

The following source data is available for figure 3:

**Source data 1.** QTL mapping results.

**Source data 2.** Phenotypes of the *DOG1* transgenic lines.

**Source data 3.** Information about Bar 1, an extra Swedish accession used as parent in one of the F2 mapping population.

**Source data 4.** SNPs and haplotypes at the *DOG1* locus for the 161 Swedish accessions used in this study.

**Source data 5.** Phenotypes and genotypes of the F2 individuals used in the QTL mapping approach.

**Source data 6.** *DOG1* PCR primers used in the complementation experiment.

variation just as well (see Materials and methods and *Supplementary file 2*), and that there may also be structural variants not present in the data we used to carry out GWAS (*Long et al., 2013*). Furthermore, it is difficult to completely rule out that these SNPs are themselves tagging rare alleles, just like SNP1 is tagging them (*Figure 3C*). Fully establishing causality would require extensive experiments using lines engineered to test all polymorphisms independently of each other, and would be almost impossible without techniques for *in situ* gene replacement given the variation in

gene expression that results from transgene insertion (*Figure 3B*). It is also crucial to distinguish between causality in the genetic sense (the focus of this paper) and molecular mechanism: we emphasize that we do not know whether these SNPs affect dormancy by regulating *DOG1*, *asDOG1* or even *CIPK20*. Given the apparent complexity of *DOG1* regulation, elucidating the causative mechanism will be a non-trivial task. It should also be noted that further alleles that we do not have the power to detect, either because they are rare or have small effect, may exist. However, while it is impossible to exclude the possibility that there are alleles we did not detect, a variance-components analysis (*Sasaki et al., 2015*) estimated that 51% of the dormancy variation is due to genetic variation in the 4.8 kb surrounding *DOG1*, which suggests that most of the variation at *DOG1* has been accounted for by the identified haplotypes (see Materials and methods). The variation in dormancy between lines carrying the same *DOG1* haplotype (*Figure 3C*) is probably due to variation at other loci (the existence of which is evident from GWAS and the F2 crosses, see *Figure 2A*, *Figure 3A* and online: https://goo.gl/30EPt3). *DOG18/RDO5/IBO*, because of its recently characterized multi-

ple natural alleles, was a good candidate to explain part of the remaining dormancy variation. However, no association was observed at this gene, presumably because the four distinct loss-of-function alleles (carried by Vår2-6, Löv-1, ÖMö2-1 and Hov3-2) segregating in our population appear to be singletons and thus cannot be detected using GWAS (and they do not segregate in our six F2 populations).

The geographic distribution of the four *DOG1* haplotypes mirrors the GR21 distribution strikingly well (*Figures 1C* and *3D* and online: https://goo.gl/3HeqX5). The majority of northern lines carry the non-dormant haplotype ND, with a small number carrying D4. Southern lines are much more diverse, with about half carrying ND, and the rest carrying either D2 or (much more rarely) D3.

To confirm that this striking variation is not an artifact of laboratory conditions, we performed field experiments in southern Sweden, utilizing two of our previous mapping populations: T880 (D2) x Grön 14 (ND) and Bar 1 (ND) x Ull2-5 (D3) (*Figure 3A*). Our goal was to track changes in *DOG1* haplotype frequency across time, from seed release to the end of the germination window. Because timing of germination partly depends on the interaction between genes and seed maturation environment (*Postma and Ågren, 2015*; *Chiang et al., 2011*; *Donohue et al., 2005*), we first grew a set of F2 individuals for each population in a common garden experiment, from germination to ripening. The progeny of these plants (F3 seeds) was then mixed and dispersed in experimental plots in early summer, using a two-block design for replication. We sampled all visible seedlings (destructive sampling) in September and November and also sampled soil shortly before winter in order to gain access to the live seeds present in newly-formed seed bank (see Materials and methods and *Supplementary file 3A*).

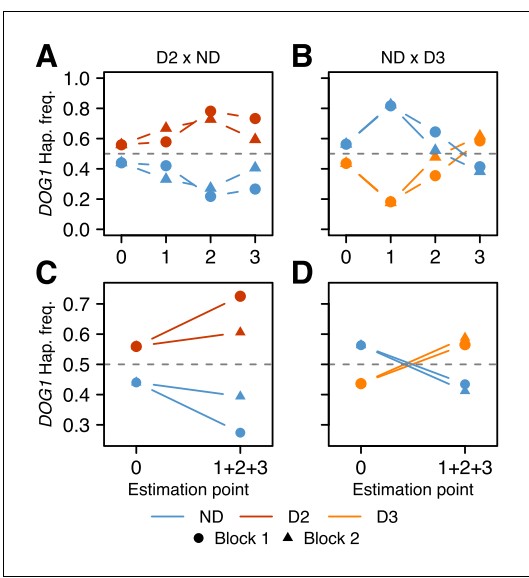

**Figure 4.** Changes in *DOG1* haplotype frequency in southern Sweden field experiments. (**A** and **C**) Frequencies in T880 (D2) x Grön 14 (ND), in which SNP2 segregates. (**B** and **D**) Frequencies in Bar 1 (ND) x Ull2-5 (D3), in which SNP3 segregates. (**A**–**B**) Frequency changes across time. Starting frequencies were estimated prior to seed dispersal and adjusted to take into account seed viability (0). Frequencies were then estimated in September (1) and November (2) by genotyping seedlings. Finally, seedlings generated from soil samples were used to estimate frequencies in the seed bank (3). (**C**–**D**) Difference between starting (0) and ending haplotype frequencies (weighted sum of 1 + 2 + 3).

The following source data and figure supplement are available for figure 4:

**Source data 1.** Genome-wide genotypes of the field experiments samples.

**Figure supplement 1.** Genome-wide changes in *DOG1* haplotype frequencies in southern Sweden field experiments.

The results were consistent across replicates, and demonstrated that the *DOG1* haplotypes can have a major effect on survival. For both populations, the frequency of the dormant haplotype increased over time (as expected given delayed germination), but the details varied greatly (*Figure 4*, *Supplementary file 3B–C*, *Figure 4—figure supplement 1* and *Figure 4—source data 1*). In the D2 x ND population, the frequency of D2 increased moderately between seed dispersal and September, and appeared to continue increasing throughout fall (it was slightly lower in the seed bank) (*Figure 4A*). This is consistent with D2 delaying germination, and delayed germination increasing seedling survival. In contrast, the frequency of D3 in the ND x D3 population decreased dramatically between seed dispersal and September, but recovered rapidly by November, indicating a dramatic delay in germination (*Figure 4B*). Furthermore, the D3 frequency in the seed bank was very high, demonstrating that this haplotype not only caused a delay in seed germination but also prevented seeds from germinating at all (*Figure 4B*).

With all live individuals at the end of the experiment counted (i.e., with the seed banks included), the proportion of dormant alleles increased massively in both populations, presumably because non-dormant seeds germinated under unfavorable conditions, and died before they could be sampled (*Figure 4C–D*). The implied selection strengths are enormous: the viability of homozygous non-dormant *DOG1* genotypes relative to homozygous dormant ones until winter was 50–60% depending on population (*Supplementary file 3D*). Of course there will be further mortality in the seed bank and not all established seedlings will survive and reproduce, but the results serve to illustrate that selection on *DOG1* could be very strong indeed, and lend credence to the notion that the observed distribution of dormancy and *DOG1* alleles (*Figures 1C* and *3D*) reflects local adaptation. Although our experiments have limitations and do not allow us to decompose total fitness, our results agree with those from previous studies, and it becomes very clear that germination timing is one of the main (if not the main) fitness components (*Postma and Ågren, 2016*; *Huang et al., 2010a*). To fully understand how selection acts will require multi-generational field experiments as selection pressures are likely to vary from year to year (and are unlikely to be at equilibrium, because of large-scale climate variations). Under the conditions of our experiment, D2 (the most common dormant haplotype) appears to have had a clear advantage over ND, whereas D3 may have been too dormant (*Figure 4*).

Furthermore, strong selection could also lead to adaptation on a very local scale. Generally speaking, deep dormancy may allow plants from the south to avoid periodic drought and high temperatures in summer, whereas northern populations may benefit from a shallower dormancy to cope with a short vegetative growth window. However, the fact that southern Sweden harbors large numbers of both dormant and non-dormant *DOG1* haplotypes demonstrates that variation for seed dormancy can be found at a very local scale, presumably due to microenvironmental variation, and the distribution of SNP4, which segregates in northern Sweden only, suggests that moderate levels of dormancy could also be beneficial at this latitude.

Finally, our results provide a dramatic illustration that the 'holy grail' of molecular ecology — finding 'the genes that matter'—may sometimes actually be obtainable (*Mitchell-Olds, 2001*; *Rockman, 2012*). That a single locus should explain most of the variation for a major life history trait is certainly surprising and arguably unprecedented. Of course we do not know how much of dormancy variation *DOG1* would explain under field conditions (and we certainly do not know how much of the variation for total fitness it would explain), but our data suggest that it is a major contributor. Although other loci also appeared to be under selection in our field experiments—as expected given that the F2 results show that other loci also contribute strongly to variation for dormancy—allele frequency changes at *DOG1* were among the strongest in both populations (*Figure 3A* and *Figure 4—figure supplement 1*). Interestingly, GWAS in a global population also identifies *DOG1* as strongly associated with flowering time variation (*Atwell et al., 2010*; *1001 Genomes Consortium 2016*), although it is not yet clear whether this association is real or spurious. If the latter, a likely explanation is that correlated selection pressures on two related life history traits—germination and flowering—has generated cross-trait genotype-phenotype correlations. If the former, *DOG1* emerges as an evolutionary 'master switch' for life history variation, perhaps similar in importance to the central flowering regulator *FLC* (*Li et al., 2014*). Indeed a recent study has demonstrated that *DOG1* may affect both dormancy and flowering (*Huo et al., 2016*), and it is interesting to note that both genes appear to be regulated through similar mechanisms, with chromatin remodelling and antisense transcripts switching off sense transcription (*Swiezewski et al., 2009*; *Footitt et al., 2015*; *Fedak et al.,*

*2016*). Perhaps such genes play general role in regulating and modulating environmentally dependent life history transitions?

In conclusion, the study of adaptive traits variation in *A. thaliana* is starting to reveal a similar picture: multiple, often complex, alleles at a relatively small number of loci account for much of the variation (*Li et al., 2014*; *Sasaki et al., 2015*; *Chao et al., 2012*; *Poormohammad Kiani et al., 2012*; *Forsberg et al., 2015*). The phenomenon is certainly not limited to *A. thaliana*—pigmentation often appears to behave similarly, for example, in humans (*Beleza et al., 2013*) as well as *Drosophila* (*Rebeiz et al., 2009*)—nor is it universal in this species (*Atwell et al., 2010*). While it is straightforward to show that local adaptation may lead to such an architecture (*Yeaman and Whitlock, 2011*), much more data—from a variety of organisms and phenotypes—are needed before we can draw general conclusions about when and why this happens.

## Material and methods

### The Swedish accessions

All the inbred lines used in this study were collected from natural populations in Sweden. Climatic and environmental conditions show extensive variation across Sweden and these populations are likely to be locally adapted. Northern Sweden is characterized by long, cold winters, mild summers and rapidly declining temperatures and day length in fall. In the south, natural populations experience a wider range of conditions than in the north, but, in general, summer is longer, warmer and dryer, while winter is shorter. The micro-environment is also very diverse is Sweden. For example, populations from the north are mainly located on eroded, south-facing rocky slopes, while the ones from the south occur in various types of habitats, such as meadows, agricultural fields, beaches and urban areas. Because of this environmental variation, the seasonal timing of key life history traits can differ between populations, although Swedish *A. thaliana* generally exhibit a winter annual life cycle. Flowering usually occurs one month later in the north (June) than in the south (May) whereas seedlings can usually be seen earlier in the north (August) than in the south (September–November) (personal observations).

All these features made the Swedish population highly suitable for studying the genetic basis of seed dormancy. The 161 lines that we used to carry out GWAS were recently sequenced and more than 4.5 millions SNPs were generated, allowing for high-resolution mapping (*Long et al., 2013*). These accessions are listed in *Figure 1—source data 1*. One line not in this set—Bar 1 (*Figure 3—source data 3*)—was used in an F2 cross and its *DOG1* genotype obtained by Sanger sequencing.

### GR21 estimation in Swedish accessions

As measuring dormancy requires fresh seeds, three biological replicates of each line were grown under long-day conditions (16 hr light, 8 hr dark) for 8 weeks at 4°C to ensure proper vernalization. Temperature was then raised to 21°C (day) and 16°C (night) until seed harvest. Watering was individually monitored and stopped when about half of the siliques of a given plant had ripened. Seeds were harvested 10 days later. After-ripening was performed in the dark at 16°C with 30% relative humidity for 21 days. Seeds from the three biological replicates were pooled together to compensate for growth chamber heterogeneity. About 100 seeds per genotype were spread on wet filter paper in Petri dishes subsequently placed in moisture chambers (*Alonso-Blanco et al., 2003*) that were located in a culture room set at 25°C with a long-day light regime. Germination rate (GR21) was determined after 7 days of incubation by scoring radicle emergence. GR21 values are available in *Figure 1—source data 1*.

### Correlations between GR21 and climate variables

We observed a strong positive correlation between GR21 and latitude in Sweden (Pearson's *r* = 0.62). Because climate varies with latitude, we assessed correlations between dormancy and various climate variables to identify those that might shape natural variation for this trait. We used climate data generated by the WorldClim project (*Hijmans et al., 2005*) as well as data gathered by *Hancock et al. (2011)*. We focused on a set of 20 variables that seemed ecologically relevant to seed dormancy (*Supplementary file 1*). We tried to minimize collinearity and redundancy when selecting variables. It should be noted that the resolution of these climatic data is really too low to

characterize accurately a small geographic area such as Sweden. Similarly to *Hancock et al. (2011)* we used a partial Mantel test (*Mantel, 1967*; *Smouse et al., 1986*) to assess the significance of the correlation between GR21 and the selected climate variables. The dependent variable was a pairwise distance matrix of the GR21 phenotype and the predictor variable was a pairwise distance matrix of the climate variables. To account for demographic history we used as covariate a pairwise kinship matrix based on the SNPs from the RegMap panel (*Horton et al., 2012*). We estimated the Mantel Spearman's coefficient using the ecodist package (*Goslee and Urban, 2007*) in the R environment (*R Core Team, 2014*). Results are presented in *Supplementary file 1*.

## Genome-wide association studies (GWAS)

To identify the genetic basis of the GR21 variation we carried out a GWAS using a mixed-model accounting for population structure (*Kang et al., 2010*; *Zhang et al., 2010*) using the SNPs generated by *Long et al. (2013)*. We identified a broad peak on chromosome five that spanned almost 50 kb and encompassed around 1800 SNPs (*Figure 2A–B*). Other peaks were found on other chromosomes but as none of them reached genome-wide significance (Bonferroni threshold, used to correct for multiple testing) they were not considered in this manuscript. The GWAS results can be viewed interactively online (https://goo.gl/30EPt3).

## Candidate genes in the associated region

Among all the genes present in the 50 kb associated region, *DOG1* was the most obvious candidate because of its major contribution to seed dormancy variation (*Alonso-Blanco et al., 2003*; *Bentsink et al., 2006*, *2010*; *Kronholm et al., 2012*). However, we also considered the possibility that genes other than *DOG1* have an effect on dormancy. We selected three candidates, based on their position (located between the significantly associated SNPs and *DOG1*) and function. One of them, *MEE62* (At5g45800), encodes a leucine-rich repeat protein kinase contributing to embryo development (*Pagnussat et al., 2005*). The two other genes, *CIPK19* (At5g45810) and *CIPK20* (At5g45820) are part of the SnRK protein family, whose members encode serine-threonine protein kinases involved in responses to salt stress and sugar and ABA signalling (*Hrabak et al. 2003*).

   We measured GR21 on insertion mutants that were obtained from the ABRC stock center (SALK_133510 - *MEE62*, SALK_044735 - *CIPK19* and GABI_533 G02 - *CIPK20*, all in Col-0 background). We used as controls the accession Col-0 (WT) and the *dog1-2* mutant—also in Col-0 background—that contains a premature stop codon in the first exon of the gene resulting in completely non-dormant seeds (*van Zanten et al., 2011*). This line was kindly provided by Wim Soppe (Max Planck Institute for Plant Breeding Research, Cologne, Germany). Germination rate after 0, 7 and 21 days of after-ripening was estimated as previously described, using three biological replicates per genotype (See Materials and methods, GR21 estimation in Swedish accessions). However, it should be noted that the vernalization treatment was not applied because all the tested genotypes are in the Col-0 background, which does not require low temperature to flower. Differences between genotypes were evaluated with a one-way ANOVA followed by a Tukey's HSD test ($p < 0.05$, n = 3).

## Linkage disequilibrium analyses in the associated region

To dissect further the main GWAS peak, we examined the linkage disequilibrium (LD) pattern in the associated region. Using custom-made R scripts (available for download at GitHub: https://goo.gl/1QdEdY) we calculated LD ($r^2$) between SNP1 and the neighboring SNPs that were spanned by the peak. As expected, we found that LD decayed within 10 kb starting from SNP1 (*Kim et al., 2007*; *Long et al., 2013*). However, a marker that we named SNP2 (c5p18592365) was still in moderate LD ($r^2 = 0.61$) with SNP1 despite a physical separation of almost 21 kb, as shown in *Figure 2—figure supplement 2*.

## GWAS with multi-locus mixed-model (MTMM)

To look for evidence of allelic heterogeneity and identify the variants potentially causing synthetic associations, we used a multi-locus mixed-model (MLMM) (*Segura et al., 2012*), which follows a stepwise approach to incorporate SNPs as covariates in the GWA model. At each step, the MLMM algorithm adds the most significant SNP to the model, before re-estimating genetic and error variances. We used the extended Bayesian information criteria (extBIC) calculated by the program to avoid

model overfitting and to select the optimal number of steps—and therefore the number of cofactors added to the model. All analysis were performed using the R version of MLMM, which was modified to be able to specify the SNP that should be used as starting point—i.e. included at the first step. By default, the program picks the most significant marker, which, in our case, would always be SNP1.

SNP3 (c5p18591702), which was identified by the MLMM program, is in complete LD ($r^2$ = 1) with two other markers. The first one, c5p18591361, was located in *DOG1* first exon and the second one, c5p18591703, in *DOG1* promoter. Any of these three markers gave similar results when included in the MLMM. In this manuscript, we are arbitrarily discussing only one of them, SNP3.

## GWAS with pairwise combinations of SNPs

A complementary way to look for evidence of allelic heterogeneity was to test the association between GR21 and pairs of markers, instead of assessing the effect of one marker at a time—as we did in the marginal GWAS. Our goal was to identify SNPs that, when combined, explained a substantial proportion of the GR21 variation. Because the GR21 phenotype has a bimodal distribution (*Figure 1A*), we assumed that variants potentially causing synthetic associations were mutually exclusive and would not cosegregate among the Swedish lines. From the SNP matrix used for GWAS (*Long et al., 2013*), we extracted 1805 SNPs that were located within the 50 kb associated region encompassing *DOG1*. To minimize the number of pairs to test, SNPs with a minor allele frequency lower than 5% were discarded. The remaining 1145 SNPs were paired (0 or 0 = 0, 1 or 0 = 1, 0 or 1 = 1, 1 or 1 = 1) to create pseudo SNPs (pSNPs) and the association between all 654,940 possible pSNPs and GR21 was tested with the GWA model used for the marginal analysis. We found that pSNP285192 (SNP1 + SNP2) and pSNP580642 (SNP2 + SNP3) shared the lowest *p*-value ($-\log_{10}p$ = 11.88) (*Supplementary file 2*). As previously mentioned, SNP3 is in strong linkage disequilibrium with two other markers ($r^2$ = 1): c5p18591361 and c5p18591703. For this reason, pSNP579096 (c5p18591361 + SNP2) and pSNP581026 (c5p18591703 + SNP2) are also among the top combinations (*Supplementary file 2*).

## Variance-components analysis

The variance-component analysis was carried out using a Python-based mixed model package 'Mixmogam' (https://github.com/bvilhjal/mixmogam, 'local_vs_global_mm' function). Phenotypic variation was decomposed into genetic variation explained by local and global genomic regions and was modelled using the following equation

$$Y = U_{local} + U_{global} + \psi$$

where $Y$ is a vector of the GR21 phenotype, $U_{local}$ and $U_{global}$ are random effects corresponding to local (*DOG1* cloned region, chr5, 18,588,990 to 18,593,792) and global relatedness (the rest of the genome) matrices, respectively, and $\psi$ is noise. IBD relatedness matrices were computed using local and global SNPs generated by *Long et al. (2013)*.

## Functional validation of *DOG1* alleles

We used two complementary approaches—QTL mapping in F2 populations and transgenic complementation—to validate the *DOG1* alleles identified in our GWAS. We generated six different F2 mapping populations in which various *DOG1* allele segregate. T880 x Grön 14, TFÄ 07 x Stenk-3 and Eden-2 x ÖMö1-7 were designed to show that the allele tagged by SNP2 was functional (all three populations) and could produce a dormant phenotype without requiring SNP1 (last two populations). Bar 1 x Ull2-5 was created to validate the allele tagged by SNP3. Finally, the two remaining populations, TAA 14 x TEDEN 03 and TEDEN 03 x TÄL 07, were used to assess the effect of SNP4. SNP and haplotype information regarding these accessions can be found in *Figure 3A* and *Figure 3—source data 3* and *4*. For each population, we grew 90 F2 individuals and estimated GR21 on their offspring (F3 seeds). F2s were grown under similar conditions as the Swedish accessions and we also followed the same protocol for seed harvest and GR21 estimation.

DNA was isolated from F2 flower buds from branches that would not produce siliques. We followed the standard protocol of the Macherey Nagel NucleoMag 96 Plant kit. Three F2 populations (Bar 1 x Ull2-5, TAA 14 x TEDEN 03 and TEDEN 03 x TÄL 07) were genotyped using a 384-SNP Illumina GoldenGate genotyping assay. Illumina's Genome Studio software and custom-made R scripts

(available for download at GitHub: https://goo.gl/1QdEdY) were used for SNP calling. The three remaining populations (T880 x Grön 14, TFÄ 07 x Stenk-3 and Eden-2 x ÖMö1-7) were genotyped by sequencing. Genomic DNA libraries were prepared using Illumina's Nextera 96 kit and sequenced on an Illumina HiSeq 2500. We aligned the resulting 125 bp paired-end reads to the *A. thaliana* TAIR 10 reference genome with the BWA-MEM algorithm (*Li, 2013*). We then followed the Genome Analysis Toolkit (GATK) recommendations for variant calling (*McKenna et al., 2010*; *DePristo et al., 2011*; *Van der Auwera et al., 2013*). Finally, custom-made R scripts (available for download at GitHub: https://goo.gl/1QdEdY) were used for SNP filtering and polarization and determine the genotype of each F2 individual in 200 kb windows, which were used as markers in downstream analyses. Raw sequencing reads were uploaded to NCBI SRA (BioProjects and BioSamples accession numbers are available in *Figure 3—source data 5*).

QTL mapping for GR21 was carried out using the 'MQM' strategy ('mqmscan()' function) implemented in the R/qtl package (*Arends et al., 2010*; *Broman et al., 2003*). The 95% LOD significance threshold was determined by running 1000 permutations per F2 population (*Figure 3—source data 1*). Genetic maps and GR21 phenotypes of the six F2 populations are available in *Figure 3—source data 5*.

For the transgenic complementation experiment, a 4.8 kb genomic fragment encompassing *DOG1* and its promoter—2.4 kb upstream start codon and 0.6 kb downstream stop codon—was amplified from DNA isolated from three different accessions carrying various *DOG1* alleles. Ale-Stenar-64–24 and Ale-Stenar-44–4 respectively carry the non-reference (dormant) allele at SNP2 and SNP3 while Eden-2 carries the reference (non-dormant) alleles at both SNPs (*Figure 3—source data 4*). Products of a Phusion-based PCR using allele-specific oligonucleotides (*Figure 3—source data 6*)—because of polymorphism at the 3' end of the gene—were cloned with the In-fusion Advantage PCR Cloning Kit (Clontech) into the EcoRI digested expression vector pGreenII 0029, which was kindly provided by Santosh Satbhai (Gregor Mendel Institute, Vienna, Austria). Constructs were transferred into *dog1–2* plants using *A. tumefaciens*. GR21 was estimated on T4 seeds descending from homozygous T3 plants grown under long-day conditions (16 hr light, 8 hr dark) at 21°C (day) and 16°C (night). Three biological replicates were grown per T3 line and their mean GR21 value was used for statistical analyses. We phenotyped a total of 12, 18 and 21 independent T3 lines for Ale-Stenar-64–24, Ale-Stenar-44–4 and Eden-2 alleles, respectively (*Figure 3—source data 2*).

## Design of field experiments

The two-year field experiments took place in Skåne, southern Sweden, between September 2013 and November 2014. The first year was dedicated to the production of the seeds that were used the following year, which consisted in a selection experiment designed to assess the adaptive relevance of seed dormancy and *DOG1* genetic variation. We focused on two different mapping populations that were previously analysed under controlled conditions (*Figure 3A*). We chose parents with contrasted origins, dormancy phenotypes and *DOG1* haplotypes. The first population (D2 x ND) was derived from the cross between T880 (southern Sweden, GR21 = 0%, haplotype D2, dormant allele at SNP2) and Grön 14 (northern Sweden, GR21 = 100%, haplotype ND, non-dormant allele at SNP2). The parents of the second population (ND x D3) were Bar 1 (northern Sweden, GR21 = 100%, haplotype ND, non-dormant allele at SNP3) and Ull2-5 (southern Sweden, GR21 = 0%, haplotype D3, dormant allele at SNP3).

Seed dormancy is heavily influenced by environmental conditions during seed maturation (*Chiang et al., 2011*; *Postma and Ågren, 2015*; *Donohue et al., 2005*). Therefore, it was necessary to use seeds produced in the field. For each population, we grew 248 F2 individuals in a common garden experiment at Ullstorp (56.066721, 13.944655), southern Sweden. Stratified seeds were sown on 6 × 11 planting trays filled with potting soil ('Lantliv Plantjord', www.rolunda.se) on 26th September 2013. Trays were watered and placed next to the planting site under a small greenhouse to maximise seed germination. Three weeks later, we transferred them into soil and extra seedlings were thinned out. Finally, trays were watered one last time to ensure a smooth transition between the greenhouse and the field. These plants grew under natural conditions during most of their life cycle and their progeny was therefore highly suitable to study seed dormancy. F3 seeds were harvested on 30th May 2014, which coincided with ripening in surrounding *A. thaliana* wild populations. Because some F2 individuals did not survive, we harvested seeds for 185 and 214 F3 families for D2 x ND and ND x D3 populations, respectively. After seed cleaning, we created pools of seeds by

mixing together 5 (D2 x ND) and 4 (ND x D3) seeds from each F3 family. We prepared a total of eight pools per population, each of them consisting of 925 and 856 F3 seeds for D2 x ND and ND x D3, respectively.

The selection experiment was conducted at Rathckegården (55.905647, 14.260095), southern Sweden, about 30 km flying distance from the seed production site. This location had previously been used to perform common garden experiments involving several *A. thaliana* lines. Consequently, the already existing seed bank prevented us from dispersing seeds directly on natural soil. To circumvent this issue, we used a mixture of potting soil ('Lantliv Plantjord', www.rolunda.se) and fine sand ('Sandlådsand', www.weibulls.com) to disperse the seeds on (ratio was 1 L of sand/10 L of soil). Two rectangular holes (220 × 100 × 5–10 cm) corresponding to both experimental blocks were dug next to each other. To avoid competition and contamination caused by other weeds growing in the vicinity of the experiment, we added a layer of landscape fabric at the bottom and on the sides of each block. This way, only gas and humidity exchanges were permitted between the ground and the artificial soil layer, excluding undesirable plant material from the experiment. Holes were then filled up with the soil mixture. Each experimental block was designed to accommodate eight 20 × 30 cm frames (four frames per population) Pools of seeds were dispersed homogeneously in their respective frames on 13th June 2014, less than 15 days after seed harvest.

## Seedlings and soil sampling

Germination usually occurs in a time window ranging from the beginning of September to mid November in *A. thaliana* populations from southern Sweden. Therefore, we harvested seedlings on 26th September 2014 (sampling point 1) and 25th November 2014 (sampling point 2) to cover the whole germination window. Because the number of seedlings was extremely low at both time points we decided to sample all visible seedlings (that were big enough to be seen, destructive sampling) instead of randomly harvesting a predefined number of plants. As a consequence, all seedlings collected in November had germinated later than those sampled in September. Additionally, a few seedlings found outside the frames—because of seed transportation—were also collected to partially compensate for the overall low number of samples.

We performed soil sampling on 26th November 2014 (sampling point 3). With the help of a little plastic scoop, we took 18 soil samples per frame, each sample covering approximately 1 of the frame surface. Therefore we covered 18 cm$^2$, which was about 3% of the total surface of a frame (600 cm$^2$). Soil samples were kept in paper envelopes placed in sealed freezing bags to make sure that soil did not dry during the transfer to Vienna. On 28th November 2014, soil samples were spread as a fine layer onto small plastic trays filled with potting soil and placed in a greenhouse under long-day conditions. The 16 trays (one for each experimental frame) were watered on regular basis and seedlings were harvested until no germinants were observed. At this point, watering was stopped and we let the trays dry out for about four weeks. After re-watering the trays, we noticed a new flush of germination that lasted for a few days. We then repeated the drought treatment two more times, until no more germination was observed. This treatment has presumably promoted the germination of most of the live seeds present in the soil samples. However, it is impossible to rule out the possibility that some live seeds did not germinate because of a too deep dormancy.

Overall, the number of samples collected at points 1, 2 and 3 was lower than expected (based on previous unpublished data). Therefore, instead of looking at allele frequency changes at the frame level—as originally planned—we decided to look at changes at the block level in subsequent analysis. *Supplementary file 3A* recapitulates the number of samples collected per block and population at the different time points.

## Genotyping of field experiments samples

We isolated DNA for all field experiments samples with the NucleoMag 96 Plant kit (Macherey Nagel) following the standard protocol. All samples were genotyped by sequencing, as previously described (See Materials and methods, Functional validation of *DOG1* alleles). Raw sequencing reads were uploaded to NCBI SRA (BioProjects and BioSamples accession numbers are available in *Figure 4—source data 1*). To bin SNPs, we again used a window size of 200 kb, which was sufficient to estimate the genome-wide allele frequency changes (*Figure 4—figure supplement 1* and *Figure 4—source data 1*). However, we used a smaller window centered on *DOG1* (22 kb, *DOG1*

ORF ±10 kb) to estimate changes at the *DOG1* locus more accurately (*Figure 4*, *Supplementary file 3B–C*).

## Estimation of genotype and allele frequency changes

Starting genotype and allele frequencies depend on the viability and the genotype of the dispersed F3 seeds. The proportion of viable seeds was determined for each F3 family by measuring the germination rate of 50–100 stratified seeds (five days, 4°C, dark, 99% humidity) as described previously (See Materials and methods, GR21 estimation in Swedish accessions). Petri dishes used to determine viabilities were kept and seven-day-old seedlings as well as imbibed ungerminated seeds were pooled together to isolate DNA. These pools were genotyped by sequencing as described previously (See Materials and methods, Functional validation of *DOG1* alleles). This allowed us to reconstruct the genotype of the parental F2s and to determine the genome-wide genotype frequencies for both populations. We were then able to estimate the genotype and allele frequencies in the pools of dispersed F3 seeds, taking into account seed viability and assuming Mendelian inheritance between generations. These frequencies were used as a proxy for starting allele frequencies at dispersal (sampling point 0).

Intermediate genotype and allele frequencies were calculated using the genotype counts observed at each sampling point (1, 2, and 3). Ending frequencies were estimated by considering all the collected samples together (sampling points 1 + 2 + 3), as a pool, regardless of their original sampling point. It should be noted that we covered only 3% of the frames surface when soil sampling. Also, we noticed that some seeds had been transported outside the frames during the course of the experiment. Therefore, the contribution of soil samples to the ending frequencies had to be adjusted accordingly (*Supplementary file 3A*). We first estimated genotype counts (*GC*) for september ($GC_1$), november ($GC_2$) and soil ($GC_3$) sampling points. We calculated ending genotype counts (*CEGC*) for each population and block using the following equation

$$CEGC = GC_1 + GC_2 + GC_3 \times area \times (1 + moved)$$

where *area* was used to compensate for the low area covered during soil sampling —It was equal to 100/3—and *moved* was the proportion of seeds that had moved outside the frames during the course of the experiment. It was calculated by dividing the number of seedlings found outside the frames at sampling points 1 and 2 by the total number of seedlings collected at both sampling points. *moved* was equal to 0.053, 0.027, 0.044 and 0.009 for D2 x ND block 1, D2 x ND block 2, ND x D3 block 1 and ND x D3 block 2, respectively. Ending genotype and allele frequencies were then derived from *CEGC*. Allele and genotype frequencies (starting, intermediate and ending) estimated for the *DOG1* locus are listed in *Supplementary file 3B–C*, respectively.

## *DOG1* genotypes performance in the field

The main focus of our field experiments was to assess differences in germination timing, and their design was not optimal for quantifying total fitness. As a consequence, we could only estimate survival between dispersal in June and the onset of winter in November. *Postma et al. (2016)* have shown that survival during this phase is critical to plant performance. Thus, we believe that our estimations of 'fitness' are realistic. The fitness contribution of the *DOG1* genotypes was quantified using starting and ending genotype frequencies at the *DOG1* locus (*Supplementary file 3C*) as well as the mean fitnesses of the tested populations (proportion of live individuals at the end of the experiment) (*Supplementary file 3A*). The marginal fitnesses $wD/D$, $wD/ND$ and $wND/ND$ of all three genotypes were modeled as follows for each block and population

$$wD/D = pp'/pp \times \bar{w}$$
$$wD/ND = pq'/pq \times \bar{w}$$
$$wND/ND = qq'/qq \times \bar{w}$$

where $pp$, $pq$ and $qq$ are the starting genotype frequencies, $pp'$, $pq'$ and $qq'$ the ending genotype frequencies and $\bar{w}$ the mean fitness. Relative fitnesses were calculated by dividing the marginal fitnesses by the marginal fitness of the fittest genotype (*Supplementary file 3D*).

## Acknowledgements

This work was supported by European Research Council grant 268962 (MAXMAP) to MN. We thank members of the Nordborg lab for illuminating discussions, Caroline Broyart, Malte Jönsson, Nils Jönsson, Svante Holm, Torbjörn Säll and Ann Espelund for help with the field experiments, Wim Soppe and Santosh Satbhai for providing us with biological material, Bjarni J Vilhjálmsson for making the 'mixmogam' package available, the VBCF NGS Unit (www.vbcf.ac.at) for performing Illumina sequencing, Harald Scheuch for running the 384-SNP Illumina GoldenGate genotyping assays, and Joy Bergelson and Caroline Dean for comments on the manuscript.

## Additional information

### Competing interests

MN: Reviewing editor, *eLife*. The other authors declare that no competing interests exist.

### Funding

| Funder | Grant reference number | Author |
| --- | --- | --- |
| European Research Council | 268962 (MAXMAP) | Magnus Nordborg |

The funders had no role in study design, data collection and interpretation, or the decision to submit the work for publication.

### Author contributions

EK, Conception and design, Acquisition of data, Analysis and interpretation of data, Drafting or revising the article; DLF, AK, ES, ÜS, Analysis and interpretation of data, Contributed unpublished essential data or reagents; VN, Acquisition of data, Contributed unpublished essential data or reagents; MN, Conception and design, Analysis and interpretation of data, Drafting or revising the article

### Author ORCIDs

Envel Kerdaffrec, http://orcid.org/0000-0001-8667-6850
Danièle L Filiault, http://orcid.org/0000-0002-2938-3071
Arthur Korte, http://orcid.org/0000-0003-0831-1463
Ümit Seren, http://orcid.org/0000-0003-4031-0428
Magnus Nordborg, http://orcid.org/0000-0001-7178-9748

## Additional files

### Supplementary files

• Supplementary file 1. Correlation between GR21 and climate variables. Partial Mantel correlations between climate variables and GR21, where the kinship between accessions is included as a covariate in the model to correct confounding caused by population structure. The Mantel *r* values are the Spearman partial correlation coefficients.

• Supplementary file 2. Local association scan using pairwise combination of SNPs. Only the top 15 combinations are displayed (see Materials and methods, GWAS with pairwise combinations of SNPs).

• Supplementary file 3. (A) Field experiments summary and sample inventory, per block and population. The number of soil samples was adjusted (Soil adj.) to take into account the size of the sampled area and seed transportation over the course of the experiment (see Materials and methods). The mean fitnesses were calculated by dividing the total number of live samples by the number of dispersed viable seeds (Total / Dispersed * Viability). (B) Summary of allele frequency changes at the *DOG1* locus, per population and block. (C) Summary of genotype frequency changes at the *DOG1*

locus, per population and block.(D) Relative fitnesses of *DOG1* alleles in field experiments, per population and block.

## Major datasets

The following datasets were generated:

| Author(s) | Year | Dataset title | Dataset URL | Database, license, and accessibility information |
|---|---|---|---|---|
| Kerdaffrec E, Filiault DL, Korte A, Sasaki E, Nizhynska V, Seren Ü, Nordborg M | 2016 | Low coverage sequencing of three *A. thaliana* F2 populations for QTL mapping | https://www.ncbi.nlm.nih.gov/bioproject/PRJNA343766 | Publicly available at NCBI SRA. |
| Kerdaffrec E, Filiault DL, Korte A, Sasaki E, Nizhynska V, Seren Ü, Nordborg M | 2016 | Low coverage sequencing of *A. thaliana* samples from Swedish field experiments | https://www.ncbi.nlm.nih.gov/bioproject/PRJNA343768 | Publicly available at NCBI SRA. |

The following previously published dataset was used:

| Author(s) | Year | Dataset title | Dataset URL | Database, license, and accessibility information |
|---|---|---|---|---|
| Long Q, Rabanal FA, Meng D, Huber CD, Farlow A, Platzer A, Zhang Q, Vilhjálmsson BJ, Korte A, Nizhynska V, Voronin V, Korte P, Sedman L, Mandáková T, Lysak MA, Hellmann I, Nordborg M | 2013 | Massive genomic variation and strong selection in *Arabidopsis thaliana* lines from Sweden | https://github.com/Gregor-Mendel-Institute/swedish-genomes | Publicly available at Github. |

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
