## [Decision Letter]

Thank you for submitting your article "Multiple alleles at a single locus control seed dormancy in Swedish *Arabidopsis*" for consideration by *eLife*. Your article has been reviewed by three peer reviewers, and the evaluation has been overseen by Christian Hardtke as the Senior Editor and Reviewing Editor.

The reviewers have discussed the reviews with one another and the Reviewing Editor has drafted this decision to help you prepare a revised submission.

Overall, the reviewers appreciate your effort to verify the importance of a locus pinpointed through GWAS in the field. There are however a number of shortcomings in the paper that have to be addressed before it can be considered for publication. Please address the reviewers' concerns as outlined below, but pay particular attention to the following points:

1) Please embed your work in the wider context and acknowledge the prior work of others in the field more comprehensively. With regard to seed dormancy in local adaptation, it appears pertinent that the work of Postma et al. (also 2016, Ann. Bot.) and Kronholm et al. 2012 is discussed in more detail. Further, please expand your Introduction and Discussion of natural seed dormancy loci, notably with regard to *DOG18/RDO5/IBO*, for which extensive natural variation and QTL have been reported (Amiguet-Vercher et al., 2015, New Phyt; Xiang et al. 2016, Plant Phys).

2) With regard to the latter, please determine whether variation in *DOG18/RDO5/IBO* exists in your populations and if so, whether it could explain some of the variation.

3) One key issue: the dormancy assays that are presented for candidate genes in the mapping interval are somewhat obsolete, because the T-DNA lines that were tested are in the rather low dormant Col-0 background. Please provide a time course of germination with shorter after-ripening intervals to exclude that none of the other loci is responsible for the phenotype.

4) Another key issue: please demonstrate that there is indeed a correlation between dormancy and *DOG1* expression levels (or possibly with the expression level of particular alternative transcripts), and that the allelic *DOG1* variants you discovered influence these expression levels. You should also take into account the very recent work by Fedak et al., 2016, PNAS, with regard to *DOG1* antisense transcripts.

*Reviewer #1:*

The case presented by Kerdaffrec et al. is very similar to that of Bentsink et al. 2006 reporting the cloning of the *DOG1* QTL. Both groups face the difficulty of identifying a gene (or several genes?) responsible to confer high dormancy in a genomic interval. It is worth therefore quickly reviewing the case of *DOG1*. Bentsink used the NILDOG17-1 line, which contains a Cvi introgression in Ler conferring a strong dormancy QTL. This Cvi introgression encompasses the same genomic interval identified by Kerdaffrec et al. Bentsink et al. mutagenized the NILDOG17-1 line with EMS to find mutants that are no longer dormant. The intrinsic assumption is that the NILDOG17-1 introgression confers high dormancy by virtue of the higher activity of a gene product. This is by no means guaranteed. It could be that one or several genes products in the NILDOG17-1 introgression have lower activity and this in turn is responsible for the higher NILDOG17-1 line dormancy. How will the EMS approach identify such gene(s)?

Instead, it seems more plausible that the EMS approach may generate mutations that lower NILDOG17-1 dormancy by disrupting the function of other dormancy genes unrelated to those truly responsible for the high dormancy conferred by NILDOG17-1. This point is illustrated by the fact that in their EMS screen Bentsink et al. reported mutations in ABA biosynthetic genes. These genes were discarded because the ABA biosynthetic genes were not in the NILDOG17-1 interval. Suppose anyway that these genes were located in the NILDOG17-1 interval. Would it necessarily mean that they are responsible for the higher dormancy conferred by the NILDOG17-1 introgression? Not necessarily.

It is therefore possible that, like ABA biosynthetic genes, *DOG1* is a gene regulating dormancy that happens to be in the NILDOG17-1 interval. Its disruption after the EMS mutagenesis would lower the dormancy of the NILDOG17-1 line without actually being the cause of the higher dormancy conferred by the non-mutagenized NILDOG17-1 line.

Besides this point I take the opportunity of this review to comment about what I perceive as peripheral problems concerning *DOG1*, which I think could also be relevant for this review. In Bentsink et al. the mutated NILDOG17-1 line was not complemented with the pBIB-*DOG1*-Cvi construct (Figure 4 of Bentsink et al. 2006). Instead Bentsink et al. transformed Ler with pBIB-*DOG1*-Cvi. The resulting transgenic plants were more dormant but strictly speaking this experiment does not qualify as a complementation analysis. Furthermore, to my best knowledge mutations in *DOG1* were only reported in NILDOG17-1 (the original *dog1* mutation) and the Col ecotype. The Col ecotype is notoriously non-dormant so evaluating the function of a dormancy gene in a non-dormant ecotype is not ideal.

So far all the evidence supporting *DOG1* is regulating dormancy is rather circumstantial, mostly coming from transgenic plants expressing *DOG1*, which leads to higher dormancy. Whether this dormancy truly illustrates *DOG1*'s function in planta or whether it is a gain-of-function, neomorphic effect of *DOG1* expression remains to be determined. It would be highly informative to know for example whether mutating *DOG1* in a pure Cvi background (e.g. by CRISPR/CAS) does significantly lower dormancy. But even if this were the case it would still not show, as discussed above, that *DOG1* is the causative agent of the higher dormancy observed in NILDOG17-1.

With these preliminary comments, here is my review of the work by Kerdaffrec et al.

Kerdaffrec et al. perform a GWAS using 161 Arabidopsis inbred lines from Sweden. Authors focus on the trait of dormancy by measuring the germination percentage of seed populations after-ripened for 21 days (GR21).

GWAS for GR21 identified a single highly significant peak covering a 50kb interval in chromosome 5 containing *DOG1*, a known dormancy gene. None of the significantly associated SNPs were located within the coding sequence of *DOG1*. Authors could not exclude that genes in the interval other than *DOG1* are involved in dormancy. Thus, GR21 values of T-DNA insertion mutants for genes present in the 50kb associated region identified by GWAS were measured (MEE62, CIPK19, CIPK20). No dormancy phenotype was found.

Comment:

As discussed above, the interval might contain genes whose product activity is negatively or positively regulating dormancy. In the former case, the lower the activity conferred by the SNPs the higher the dormancy whereas in the latter case the higher the activity conferred by the SNPs the higher the dormancy. A T-DNA mutant in a negative and positive regulator of dormancy will increase and decrease dormancy, respectively. Therefore, a T-DNA mutant analysis must ensure that both cases can be found. Under the conditions used, Col-0 control seeds are no longer dormant after 21 days of after-ripening. This is illustrated by the fact that no difference in dormancy can be detected between *dog1-2* and Col.

With long after-ripening times only high dormancy phenotypes in the insertion lines can be potentially detected. Furthermore, there is even the risk of missing mild higher dormancy phenotypes.

Authors should shorten their after-ripening time in order to detect dormancy in the Col-0 control in order to see whether they can detect low or high dormancy in their insertion lines. Also, many other T-DNA insertions in other genes present in the interval should be tested.

Given the presence of *DOG1* in the interval authors propose that allelic heterogeneity at *DOG1* gives rise to synthetic or "ghost" associations. Authors note the presence of SNP2 in the promoter region of *DOG1*. Furthermore, authors reason that since *DOG1* expression strongly correlates with seed dormancy (Chiang et al. 2011; Nakabayashi et al. 2012), SNP2 could contribute to GR21 variation.

Comment:

One should exert caution about the claim that *DOG1* expression is strongly correlating with seed dormancy. Available literature shows that this correlation is sometimes observed (as in the case of Ler and Cvi, which was discussed in the original paper identifying *DOG1* (Bentsink et al. 2006) but also sometimes not observed. For example, in Figure 6 of Bentsink et al. one can see that *DOG1* expression in Col-0 (a notoriously non-dormant ecotype) is much higher than Ler and, critically, higher than that in Cvi (a notoriously dormant ecotype). Furthermore, the *DOG1* expression in Col-0 is comparable to that of NILDOG17-1 (the introgression line used in the paper to map *DOG1*). If I remember well, Bentsink et al. do not comment about this nor do they discuss whether the dormancy levels of the other ecotypes studied in Figure 6 correlate with *DOG1* expression levels.

Concerning the reference Chiang et al. 2011, authors establish a correlation in the expression of DOG1 among in Arabidopsis lines whose geographical distances and latitudes are much larger (southern Europe vs. northern Europe) than in this study. Unlike the present study, the Chiang et al. study also examines the particular case of the response of *DOG1* expression to changes in environmental factors, such as temperature, during seed development, which affect final seed dormancy levels.

The Chiang et al. study uses three Swedish lines, Lov-5 (Lv), Bil-7 (Bi) and Ull2-3 (Ul). Lv and Bi live in similar latitudes, which appear to be equivalent to those of the northern Swedish lines used by Kerdaffrec et al. The Ul line lives in a latitude equivalent to that of the southern Swedish lines used by Kerdaffrec et al. Nevertheless, these three lines belong to the northern group in Chiang et al. and they do not appear to report marked differences regarding DOG1 expression between these three accessions.

The Nakabayashi et al. study is mainly concerned with the Ler and Cvi ecotypes where a correlation between *DOG1* expression and dormancy levels takes place. However, in a more recent study Nakabayashi et al. take notice of the absence of correlation between DOG1 expression and dormancy levels in Col-0 ("Surprisingly, a comparison of DOG1 transcript and dormancy levels in the low dormant accessions Ler and Col, and the dormant line NIL *DOG1* (which contains the Cvi allele of *DOG1*) showed that this correlation was absent in Col, which has relatively high *DOG1* transcript levels (Figure 5A and 5B)"). Furthermore, in that study *DOG1* protein levels in Col, Ler and Cvi also correlate with *DOG1* mRNA levels. This clearly poses a challenge to the notion that *DOG1* expression and dormancy levels are correlated. This led Nakabayashi et al. to study *DOG1* protein self-binding ability, which could be associated with a change in its activity. Self-binding ability would vary according to the different *DOG1* isoforms produced by splicing variants of the *DOG1* gene (Nakabayashi et al. 2015). Finally, Nakabayashi et al. propose that *DOG1* self-binding ability can contribute to natural variation (Nakabayashi et al. 2015).

Because of the significant uncertainties regarding the link between *DOG1* expression and dormancy levels, authors must show that there is indeed a correlation between the GR21 values and the *DOG1* expression in the dormancy of the Swedish lines particularly those carrying SNP2 and SNP3.

After using different SNPs as starting points for a multi-locus GWAS, authors found that the optimal model with SNPS located in the promoter region of *DOG1* (SNP2 and SNP3) explains 57% of the variations in dormancy among the Swedish Arabidopsis inbred lines.

Comment:

This reviewer wonders about the contribution of the remaining SNPs. There are some SNPs in chromosomes 3 but also chromosome 4 nearly reaching the threshold of significance. Could they represent potential loci regulating dormancy?

Next authors use a transgenic approach to evaluate the contribution of the dormant alleles (SNP2 or SNP3) using the *dog1-2* mutant background. Transgenes containing SNP2 and SNP3 confer higher dormancy than controls (Figure 3).

Comment:

In absence of quantification of *DOG1* mRNA levels, this approach is not really testing the hypothesis that SNP2 and SNP3 contribute to dormancy by virtue of their effect of *DOG1* gene expression.

As above, authors must show a correlation between *DOG1* mRNA levels and dormancy in the three transgenic lines of Figure 3.

*Reviewer #2:*

In this manuscript the authors use genome-wide association (GWA) mapping in a set of 161 accessions from Sweden to identify the genetic architecture underlying variation in seed dormancy in *Arabidopsis thaliana*. The reported GWA mapping results identify *DOG1* as a strong candidate locus accounting for 57% of the variation. This confirms the previous results that identify *DOG1* as the causal gene underlying a seed dormancy QTL in a diallelic synthetic population, correctly cited in this manuscript. Using 8 different diallelic synthetic F2 populations, and transgenic complementation, the authors go on to identify 4 *DOG1* haplotypes in this Swedish population (3 dormancy and 1 non-dormancy alleles). Interestingly, the distribution of these alleles on the landscape matches the distribution of the dormancy phenotype. The non-dormancy allele is mainly found in the northern plants where non-dormancy is prevalent. The 2 most prevalent dormancy alleles being found in the southern plants where dormancy is more common. This suggests that variation at *DOG1* is adaptive, and this supported previously published work, correctly cited in this manuscript. To obtain evidence that these various *DOG1* alleles are adaptive the authors performed a 1-year, single site common garden experiment in the field in Southern Sweden. At this site the authors evaluate the allele frequency of the non-dormant allele and the 2 most common dormancy alleles that are segregating in 2 diallelic synthetic F2 populations. By assessing the allele frequency at sowing in early spring and again in September, November and in the seed bank the authors reveal that the frequency of both *DOG1* dormancy alleles increases from sowing to seed back and that the frequency of the non-dormancy *DOG1* allele decreases. The authors conclude that this evidence 'lends credence to the notion that the observed distribution of dormancy and *DOG1* alleles reflects local adaptation'. I think this is a very reasonable statement. It is of cause a shame that the authors did not repeat this experiment over multiple years and also included a northern site where they could have tested the adaptive potential of the non-dormancy allele (early germination to avoid early killing frosts in the fall perhaps).

In conclusion, the authors make the statement 'the study of flowering and dormancy variation in *A. thaliana* are starting to reveal a similar picture: multiple, often complex, alleles at a relatively small number of loci account for much of the variation (Sasaki et al. 2015; Li et al. 2014)'. The authors could have also made a similar statement about variation in mineral uptake by plants (Chao et al., 2012; Forsberg et al., 2015).

Overall this is a nice GWAS which gets to the underlying alleles through a combination of association genetics, linkage genetics in F2 mapping populations and transgenic complementation. This in itself is not novel in the GWAS field. However, what makes this manuscript stand out for me is the field work component that starts to test in the field if the different alleles of *DOG1* are adaptive. These field experiments have limitations (only performed in 1 year and at one site) but I think these experiments still differentiate this work from the growing body of published GWAS, and as such raise the potential impact of this work.

*Reviewer #3:*

The manuscript provides interesting results on the genetic basis and fitness consequences of dormancy variation in *A. thaliana*. A significant finding is that in Swedish material, one locus, *DOG1*, is highly important in governing variation in dormancy. Earlier authors have also shown that *DOG1* is likely important for local adaptation, but with different sampling and methods. Further, the authors have conducted field experiments to evaluate the fitness consequences of the variation, a valuable contribution. However, this part of the manuscript would benefit from clarification.

The genetics part provides interesting results. The complexities of mapping are nicely illustrated and resolved in this case study. Combining QTL mapping, association mapping, and transformations seems useful to characterize the effects of different haplotypes.

The data on correlations with climate variables do not seem so informative. With sampling heavily in the south and then in the north, with little in between, most variables are highly correlated with latitude. In fact, a more interesting question would be whether any climate variables can account for the considerable variation in GR21 in southern Sweden. For this reason, it would also be interesting to see on the map where the field experiment was located.

As mentioned, the field experiment is potentially very interesting. In the main text, the experiment is not sufficiently described. While the details should of course be in Materials and methods, it would be nice to have enough in the results to be able to follow the reasoning. For instance, the final sample sizes should be listed, and the origin of the materials, and the place of the field site should be on the map.

Also, it was not really clear what populations the allele frequencies that are shown were calculated for. Seeds were sown, they germinated, and were sampled in September and November and later. Reading details in Materials and methods shows that the allele frequencies are of seedlings available at the sampling date, and if I assume correctly, the seedlings were discarded after sampling. In this case, the allele frequencies are not for population data, but for germinating seeds. Any differences e.g. in post-germination survival are not included? So seeds that inherited for parents a late germinating haplotype also germinate later in the wild, and other fitness components were not studied. Now the non-dormant populations are all from the north – the poor success of the non-dormant seeds/seedlings could presumably be related to other linked loci in the F2 progenies. The delay of germination of the D2 and D3 haplotypes seemed pretty similar earlier (in the transformants). At least in some parts of southern Sweden nearly a third of the plants seem to be very early germinating, and the frequencies non-dormant haplotypes are very high. So, one could assume that in southern Sweden there are other important fitness components. If there is something I misunderstood, the writing could be clearer, if not, then it would be good to point out that only a part of the life cycle was studied and that there may be other fitness effects later. This part of the work could be much clarified. (Further, to demonstrate local adaptation, it would of course have been good to have another experiment in the north.)

Kronholm et al. (2012) did a thorough study on *DOG1* effects, in wider geographical area. It would seem that the present work could be more compared to those results. The key finding of multiple haplotypes was already presented. They also had extensive analysis of the phenotypic variation. Now the paper is just referenced a couple of times.

---

## [Author Response]

*Overall, the reviewers appreciate your effort to verify the importance of a locus pinpointed through GWAS in the field. There are however a number of shortcomings in the paper that have to be addressed before it can be considered for publication. Please address the reviewers' concerns as outlined below, but pay particular attention to the following points:*

*1) Please embed your work in the wider context and acknowledge the prior work of others in the field more comprehensively. With regard to seed dormancy in local adaptation, it appears pertinent that the work of Postma et al. (also 2016, Ann. Bot.) and Kronholm et al. 2012 is discussed in more detail. Further, please expand your Introduction and Discussion of natural seed dormancy loci, notably with regard to DOG18/RDO5/IBO, for which extensive natural variation and QTL have been reported (Amiguet-Vercher et al., 2015, New Phyt; Xiang et al. 2016, Plant Phys).*

We have expanded the Introduction to describe previous work better, and also added references throughout. We discuss the diverse mechanisms apparently at play in *DOG1* regulation and the existence of natural variation at other loci than *DOG1*. The work of Postma et al. 2016 is highlighted and discussed, and we emphasize the fact that allelic heterogeneity at the *DOG1* locus has already been extensively characterised by Kronholm et al. (2012).

However, because our manuscript aims at studying the genetic architecture of a complex trait rather than characterizing the molecular mechanisms involved, we keep the description of the molecular aspects of seed dormancy brief.

*2) With regard to the latter, please determine whether variation in DOG18/RDO5/IBO exists in your populations and if so, whether it could explain some of the variation.*

Xiang et al. (2016) analysed genetic variation at the *DOG18/RDO5/IBO* locus in 855 accessions from the 1001 genomes project. They characterized 180 accessions from Sweden (161 of which form our GWAS panel) and predicted that 5 lines (Vår2-6, Hov3-2, Tny-04, Löv-1 and Omö2-1) carry private loss-of-function *DOG18/RDO5/IBO* alleles. We could not observe any association at this locus in our GWAS – as expected because the described loss-of-function alleles are singletons. Moreover, we did not find any QTLs overlapping the *DOG18/RDO5/IBO* locus in our linkage mapping analyses, which was again expected since none of the described alleles segregate in our six F2 populations. In conclusion, *DOG18/RDO5/IBO* alleles probably do not have a significant effect on dormancy in Sweden. This is now explicitly stated.

*3) One key issue: the dormancy assays that are presented for candidate genes in the mapping interval are somewhat obsolete, because the T-DNA lines that were tested are in the rather low dormant Col-0 background. Please provide a time course of germination with shorter after-ripening intervals to exclude that none of the other loci is responsible for the phenotype.*

This is a very perceptive comment. We were aware of the problem and realized it might be a concern some time ago. For this reason, we have been performing the missing experiment over the course of the last several months. We measured the germination rate of the T-DNA lines after 0, 7 and 21 days of after-ripening. Interestingly, without after ripening, the *cipk20* mutant was less dormant than Col-0 but still more dormant than the *dog1-2* mutant (see Figure 5). After 7 days, although *cipk20* had increased its germination rate, it remained more dormant than the other tested lines. Finally, no difference was observed after 21 days, as already reported in the submitted version of the manuscript. These results are now presented in Figure 2—figure supplement 2 and mentioned in the text.

Author response image 1.**DOI:**
http://dx.doi.org/10.7554/eLife.22502.021

We have no idea what the mechanism behind these differences are. *CIPK20* may be directly involved in regulating dormancy, or it may simply be that the *cipk20* insertion affects *DOG1* regulation. Conversely, we cannot be sure that the SNPs we identify do not affect *CIPK20* (although if they do, the effect must be trans-acting, because of our transgenic results).

Crucially, none of this really affects our paper. Our transgenic complementation experiments unambiguously show that *DOG1* allelic variation explains part of the dormancy variation, and this fact strongly supports our GWAS results, which point towards a massive role for this allelic variation in explain the population variation for dormancy. We cannot – and do not – say anything about the molecular mechanisms by which this variation causes dormancy variation.

Elucidating these mechanisms is obviously very interesting, but is very much out of the scope of this paper. Indeed, as anyone who has followed the *FLC* story will know, it will likely require many years and many papers. We now explicitly state this in the paper.

*4) Another key issue: please demonstrate that there is indeed a correlation between dormancy and DOG1 expression levels (or possibly with the expression level of particular alternative transcripts), and that the allelic DOG1 variants you discovered influence these expression levels. You should also take into account the very recent work by Fedak et al., 2016, PNAS, with regard to DOG1 antisense transcripts.*

As pointed out by the reviewers, the correlation between *DOG1* expression and seed dormancy is not always obvious and we have modified the manuscript accordingly. The only reason we mentioned this correlation is because it helped us clarify the GWAS results. We found that one of the associated SNPs (SNP2) was located in the promoter of *DOG1* and because *DOG1* expression can be a good predictor of dormancy, we reasoned that this SNP could contribute to dormancy variation. Later, the functional validation experiments confirmed that the allele tagged by SNP2 (and SNP3, also in the promoter) was indeed causative (Figure 3). It seems plausible that this causation does indeed somehow involve regulation of *DOG1* expression (and we now mention Fedak et al., 2016), but we make no such claims, and investigating this is beyond the scope of the paper.

*Reviewer #1:*

*The case presented by Kerdaffrec et al. is very similar to that of Bentsink et al. 2006 reporting the cloning of the DOG1 QTL. Both groups face the difficulty of identifying a gene (or several genes?) responsible to confer high dormancy in a genomic interval. It is worth therefore quickly reviewing the case of DOG1. Bentsink used the NILDOG17-1 line, which contains a Cvi introgression in Ler conferring a strong dormancy QTL. This Cvi introgression encompasses the same genomic interval identified by Kerdaffrec et al. Bentsink et al. mutagenized the NILDOG17-1 line with EMS to find mutants that are no longer dormant. The intrinsic assumption is that the NILDOG17-1 introgression confers high dormancy by virtue of the higher activity of a gene product. This is by no means guaranteed. It could be that one or several genes products in the NILDOG17-1 introgression have lower activity and this in turn is responsible for the higher NILDOG17-1 line dormancy. How will the EMS approach identify such gene(s)?*

[…]

*Comment:*

As discussed above, the interval might contain genes whose product activity is negatively or positively regulating dormancy. In the former case, the lower the activity conferred by the SNPs the higher the dormancy whereas in the latter case the higher the activity conferred by the SNPs the higher the dormancy. A T-DNA mutant in a negative and positive regulator of dormancy will increase and decrease dormancy, respectively. Therefore, a T-DNA mutant analysis must ensure that both cases can be found. Under the conditions used, Col-0 control seeds are no longer dormant after 21 days of after-ripening. This is illustrated by the fact that no difference in dormancy can be detected between dog1-2 and Col.

*With long after-ripening times only high dormancy phenotypes in the insertion lines can be potentially detected. Furthermore, there is even the risk of missing mild higher dormancy phenotypes.*

*Authors should shorten their after-ripening time in order to detect dormancy in the Col-0 control in order to see whether they can detect low or high dormancy in their insertion lines. Also, many other T-DNA insertions in other genes present in the interval should be tested.*

We agree with this point. We have foreseen this potential issue and performed a much better experiment, the outcome of which is discussed above (as well as in the manuscript). Generally speaking, we do not think that strong conclusions can be drawn from t-DNA analysis, especially in the context of studying natural variation. Regarding our associated region, we note that Bentsink et al. (2006) have tested insertion lines for many of the genes present in the NILDOG17-1 Cvi introgression and only reported the effect of *DOG1*.

*Given the presence of DOG1 in the interval authors propose that allelic heterogeneity at DOG1 gives rise to synthetic or "ghost" associations. Authors note the presence of SNP2 in the promoter region of DOG1. Furthermore, authors reason that since DOG1 expression strongly correlates with seed dormancy (Chiang et al. 2011; Nakabayashi et al. 2012), SNP2 could contribute to GR21 variation.*

[…]

*Because of the significant uncertainties regarding the link between DOG1 expression and dormancy levels, authors must show that there is indeed a correlation between the GR21 values and the DOG1 expression in the dormancy of the Swedish lines particularly those carrying SNP2 and SNP3.*

We respond to this comment above.

*After using different SNPs as starting points for a multi-locus GWAS, authors found that the optimal model with SNPS located in the promoter region of DOG1 (SNP2 and SNP3) explains 57% of the variations in dormancy among the Swedish Arabidopsis inbred lines.*

*Comment:*

*This reviewer wonders about the contribution of the remaining SNPs. There are some SNPs in chromosomes 3 but also chromosome 4 nearly reaching the threshold of significance. Could they represent potential loci regulating dormancy?*

Three SNPs at the *DOG1* locus (SNP2-SNP4) explain about 57% of the variation. Most of the remaining 43% is likely due to other loci (the SNP-heritability is nearly 1), some of which could well be tagged by our non-significant association. Because we could not identify any obvious candidate genes in the vicinity of these, we chose to focus on the strong association at the *DOG1* locus only (the absence of any association at *RDO5* is now explicitly discussed). The results are freely browsable online for anyone interested in dormancy variation.

*Next authors use a transgenic approach to evaluate the contribution of the dormant alleles (SNP2 or SNP3) using the dog1-2 mutant background. Transgenes containing SNP2 and SNP3 confer higher dormancy than controls (Figure 3).*

*Comment:*

*In absence of quantification of DOG1 mRNA levels, this approach is not really testing the hypothesis that SNP2 and SNP3 contribute to dormancy by virtue of their effect of DOG1 gene expression.*

*As above, authors must show a correlation between DOG1 mRNA levels and dormancy in the three transgenic lines of Figure 3.*

It is true that the transgenic approach does not test whether SNP2 and SNP3 regulate *DOG1* expression. However, as explained in the response to the main points above, this manuscript does not aim at testing this hypothesis. The transgenic experiment was solely designed to demonstrate that the distinct *DOG1* alleles tagged by SNP2 and SNP3 are functional without making any claims about the underlying mechanism.

*Reviewer #2:*

*In this manuscript the authors use genome-wide association (GWA) mapping in a set of 161 accessions from Sweden to identify the genetic architecture underlying variation in seed dormancy in Arabidopsis thaliana. The reported GWA mapping results identify DOG1 as a strong candidate locus accounting for 57% of the variation. This confirms the previous results that identify DOG1 as the causal gene underlying a seed dormancy QTL in a diallelic synthetic population, correctly cited in this manuscript. Using 8 different diallelic synthetic F2 populations, and transgenic complementation, the authors go on to identify 4 DOG1 haplotypes in this Swedish population (3 dormancy and 1 non-dormancy alleles). Interestingly, the distribution of these alleles on the landscape matches the distribution of the dormancy phenotype. The non-dormancy allele is mainly found in the northern plants where non-dormancy is prevalent. The 2 most prevalent dormancy alleles being found in the southern plants where dormancy is more common. This suggests that variation at DOG1 is adaptive, and this supported previously published work, correctly cited in this manuscript. To obtain evidence that these various DOG1 alleles are adaptive the authors performed a 1-year, single site common garden experiment in the field in Southern Sweden. At this site the authors evaluate the allele frequency of the non-dormant allele and the 2 most common dormancy alleles that are segregating in 2 diallelic synthetic F2 populations. By assessing the allele frequency at sowing in early spring and again in September, November and in the seed bank the authors reveal that the frequency of both DOG1 dormancy alleles increases from sowing to seed back and that the frequency of the non-dormancy DOG1 allele decreases. The authors conclude that this evidence 'lends credence to the notion that the observed distribution of dormancy and DOG1 alleles reflects local adaptation'. I think this is a very reasonable statement. It is of cause a shame that the authors did not repeat this experiment over multiple years and also included a northern site where they could have tested the adaptive potential of the non-dormancy allele (early germination to avoid early killing frosts in the fall perhaps).*

It is indeed pity that we could not perform a similar experiment in the north. It was originally planned, but cancelled for logistic reasons.

*In conclusion, the authors make the statement 'the study of flowering and dormancy variation in A. thaliana are starting to reveal a similar picture: multiple, often complex, alleles at a relatively small number of loci account for much of the variation (Sasaki et al. 2015; Li et al. 2014)'. The authors could have also made a similar statement about variation in mineral uptake by plants (Chao et al., 2012; Forsberg et al., 2015).*

We have added them to the main text.

*Overall this is a nice GWAS which gets to the underlying alleles through a combination of association genetics, linkage genetics in F2 mapping populations and transgenic complementation. This in itself is not novel in the GWAS field. However, what makes this manuscript stand out for me is the field work component that starts to test in the field if the different alleles of DOG1 are adaptive. These field experiments have limitations (only performed in 1 year and at one site) but I think these experiments still differentiate this work from the growing body of published GWAS, and as such raise the potential impact of this work.*

Thank you for the fair comments.

*Reviewer #3:*

*The manuscript provides interesting results on the genetic basis and fitness consequences of dormancy variation in A. thaliana. A significant finding is that in Swedish material, one locus, DOG1, is highly important in governing variation in dormancy. Earlier authors have also shown that DOG1 is likely important for local adaptation, but with different sampling and methods. Further, the authors have conducted field experiments to evaluate the fitness consequences of the variation, a valuable contribution. However, this part of the manuscript would benefit from clarification.*

*The genetics part provides interesting results. The complexities of mapping are nicely illustrated and resolved in this case study. Combining QTL mapping, association mapping, and transformations seems useful to characterize the effects of different haplotypes.*

*The data on correlations with climate variables do not seem so informative. With sampling heavily in the south and then in the north, with little in between, most variables are highly correlated with latitude. In fact, a more interesting question would be whether any climate variables can account for the considerable variation in GR21 in southern Sweden.*

We agree that the correlations with climate variables are not so informative as they are highly confounded by latitude. The resolution of the available climate data is not precise enough to characterize southern Sweden accurately. However, we also found that temperature is the best predictor of dormancy in southern Sweden. It could for example reflect the difference between urban / rural environments, as most of the lines sampled from densely inhabited locations are very dormant. However, because of the limited resolution, these observations should of course be taken with a grain of salt. The fact that we observe such an extensive dormancy variation in southern Sweden probably indicates that microenvironmental variation (soil properties, human or animal disturbance etc.) could play an important role. Unfortunately, we do not have good enough data to address this question.

*For this reason, it would also be interesting to see on the map where the field experiment was located.*

We provide names and GPS coordinates of the field sites in the Methods section. We also added the location of the field experiments to the online version of the maps.

*As mentioned, the field experiment is potentially very interesting. In the main text, the experiment is not sufficiently described. While the details should of course be in Materials and methods, it would be nice to have enough in the results to be able to follow the reasoning. For instance, the final sample sizes should be listed, and the origin of the materials, and the place of the field site should be on the map.*

Although this is a valid point, it is hard to list the final sample sizes in the main text (that would be too many numbers). Instead, we chose to present them in a table ([Supplementary-material SD11-data]), which unfortunately, cannot be attached to Figure 4. However, we added a reference to this table in the paragraph describing the field experiments. As previously mentioned, the location of the experiments has been added to the online maps.

*Also, it was not really clear what populations the allele frequencies that are shown were calculated for. Seeds were sown, they germinated, and were sampled in September and November and later. Reading details in Materials and methods shows that the allele frequencies are of seedlings available at the sampling date, and if I assume correctly, the seedlings were discarded after sampling. In this case, the allele frequencies are not for population data, but for germinating seeds.*

We show allele frequencies for all the live individuals that were collected during the course of the experiment. These are all the visible seedlings (big enough to be seen, cotyledon stage) and the seeds from the seed bank that we manage to make germinate. It was indeed a destructive sampling, as we collected the whole plants. This way, we were sure that the seedlings collected in November had germinated later than those collected in September. Therefore, it is true that the allele frequencies presented are only for seeds that germinated and turned into a seedling. This has now been clarified in the main text and Methods.

*Any differences e.g. in post-germination survival are not included? So seeds that inherited for parents a late germinating haplotype also germinate later in the wild, and other fitness components were not studied.*

The experiment was originally conceived to show that the variation at *DOG1* could affect germination timing, hence the deliberate choice of randomly dispersing mixture of seeds. Such an experimental design is not optimal to study post germination events and fitness consequences, as it would have been hard to keep track of individual plants.

Postma et al. (2016), with a different – and complementary – strategy, could show that the transition between seed to seedling is the most critical phase the plant life cycle. This is now explicitly discussed.

*Now the non-dormant populations are all from the north – the poor success of the non-dormant seeds/seedlings could presumably be related to other linked loci in the F2 progenies.*

As mentioned at the end of the field experiments part, we also observed changes in allele frequencies at other loci, in both populations (see Figure 4—figure supplement 1). It is thus very likely that loci other than *DOG1* could have been under selection in our experiments and contributed to the performance of the seeds. While we cannot rule out the effect of tightly linked loci, we note that allele frequency changes at DOG1 are massive relative to the rest of the genome.

*The delay of germination of the D2 and D3 haplotypes seemed pretty similar earlier (in the transformants). At least in some parts of southern Sweden nearly a third of the plants seem to be very early germinating, and the frequencies non-dormant haplotypes are very high. So, one could assume that in southern Sweden there are other important fitness components.*

As mentioned in the Discussion, the high frequency of the ND haplotype in the south could reflect the fact that adaptation might happen at a very small scale, presumably because of microenvironmental differences.

If there is something I misunderstood, the writing could be clearer, if not, then it would be good to point out that only a part of the life cycle was studied and that there may be other fitness effects later.

We edited the manuscript (Discussion and Methods) to highlight the fact that we only focus on the early stages of the life cycle.

*This part of the work could be much clarified. (Further, to demonstrate local adaptation, it would of course have been good to have another experiment in the north.)*

We completely agree with the fact that having a second experiment in the north would have been good. It was originally planned, but cancelled for logistic reasons.

*Kronholm et al. (2012) did a thorough study on DOG1 effects, in wider geographical area. It would seem that the present work could be more compared to those results. The key finding of multiple haplotypes was already presented. They also had extensive analysis of the phenotypic variation. Now the paper is just referenced a couple of times.*

We agree, and now explicitly mention this. The main difference lies in the fact that we show a massive effect in a local population.